# Toward Accelerated Training of Parallel Support Vector Machines Based on Voronoi Diagrams

**DOI:** 10.3390/e23121605

**Published:** 2021-11-29

**Authors:** Cesar Alfaro, Javier Gomez, Javier M. Moguerza, Javier Castillo, Jose I. Martinez

**Affiliations:** Department of Computer Science, University Rey Juan Carlos, 28933 Móstoles, Spain; cesar.alfaro@urjc.es (C.A.); javier.moguerza@urjc.es (J.M.M.); javier.castillo@urjc.es (J.C.); joseignacio.martinez@urjc.es (J.I.M.)

**Keywords:** classification, machine learning, Support Vector Machines, sensor networks, distributed algorithms

## Abstract

Typical applications of wireless sensor networks (WSN), such as in Industry 4.0 and smart cities, involves acquiring and processing large amounts of data in federated systems. Important challenges arise for machine learning algorithms in this scenario, such as reducing energy consumption and minimizing data exchange between devices in different zones. This paper introduces a novel method for accelerated training of parallel Support Vector Machines (pSVMs), based on ensembles, tailored to these kinds of problems. To achieve this, the training set is split into several Voronoi regions. These regions are small enough to permit faster parallel training of SVMs, reducing computational payload. Results from experiments comparing the proposed method with a single SVM and a standard ensemble of SVMs demonstrate that this approach can provide comparable performance while limiting the number of regions required to solve classification tasks. These advantages facilitate the development of energy-efficient policies in WSN.

## 1. Introduction

Machine learning applications are radically changing our world as a key asset of an Information Society. New algorithms and methods for data processing and analysis, along with the capacity to deal with large and complex datasets, has led to the rise of a new industry. Over the next decades, data science and machine learning are expected to transform the way in which we interact with our surrounding environment.

One of the main challenges is to effectively prepare and analyze vast and distributed datasets. Classical algorithms for classification, such as convolutional neural networks (CNN) [1,2] or SVMs [3], are being pushed to their limits. Therefore, it is essential to develop efficient parallel architectures and techniques that can cope with massive data in distributed systems. As a result, algorithm parallelization is taking a key role, as it enables the exploitation of computing power available in large data centers, especially in cloud computing environments, to train and deploy these algorithms.

More classical machine learning algorithms, such as SVMs, can also be used as a viable alternative for classification of large datasets. Nonetheless, one of their main disadvantages is that, unlike CNN and other intrinsically parallel algorithms, SVMs lack from such property. For this reason, several proposals have been presented for their parallelization [4,5,6]. In general, parallel Support Vector Machines (pSVMs) are based on algorithm modifications to execute some code sections simultaneously. As well, alternative approaches consider incremental executions deployed on distributed architectures, such as MapReduce [7].

In this article, we present an alternative method for machine learning classification via SVMs, specially designed for structures similar to a federated network of sensors, such as wireless sensor networks [8]. These networks are characterized by the fact that it is necessary to discern between two classes in each region. These tasks arise in many contexts, such as decentralized intrusion detection systems [9], controlling environmental conditions in smart buildings [10], or emergency alert networks [11], among others. Processing large datasets acquired by WSN devices can be challenging. Specific goals in this context are to minimize communication between nodes or groups of nodes, to optimize energy consumption, as well as to attain conservative management of limited storage capacity [12].

The main contribution of our algorithm, in comparison to other similar approaches, is that it takes advantage of this kind of spatial distribution. Roughly speaking, our method works as a guided ensemble-type approach. In practice, this spatial distribution can be emulated by dividing the dataset into Voronoi regions [13]. In the case of sensor networks, data subregions contain almost complete Voronoi regions, with a rather empty intersection to other regions. At this point, it is important to remark that, in cases in which the spatial distribution of the data is known in advance and made up of small groups, this process could be avoided by using the known groups as approximated Voronoi-type regions. However, under the presence of large groups of data, the use of Voronoi regions will still be of help from a computational point of view. For the sake of completeness, in this paper, we describe the full process of building the Voronoi regions although, as commented above, it could be skipped in some situations.

In the same way, each task can be independently solved, using a standard SVM implementation such as libSVM [14], already available in popular programming languages such as Python, R, or C. As a result, any system already based on SVM can take advantage of this method, not only to reduce execution time but to also increase its processing capacity.

To this aim, we create a set of small SVMs that work as an ensemble of classifiers [15]. The key point is that members of the ensemble can be trained following a parallelization scheme. The success of these kinds of ensembles based on SVMs have already been proved in [16]. In related work, however, the SVM used for the selection of the ensemble does not admit parallelization.

The rest of the paper is organized as follows. We review previous related work in Section 2. In Section 3, the proposed algorithm is presented. Then, Section 4 describes the experimental setup to validate the proposed algorithm and presents the discussion of the results. Finally, the main conclusions and future lines of work are presented in Section 5.

## 2. Related Works

Nowadays, machine learning algorithms play a central role in wireless sensor networks [17]. In particular, SVMs are involved in diverse applications in this context such as localization techniques, anomaly and fault detection, or congestion control, among others. The new method introduced in this paper is based on the parallel implementation of SVM algorithms in Voronoi regions, efficiently combining selected results from some of these regions, following ensemble learning principles. In this section, we review the main background machine learning concepts and tools related to this work.

### 2.1. Support Vector Machines

SVMs are one of the most popular supervised learning methods that is used for both classification and regression tasks. They appeared by the end of the last century as optimal margin classifiers in the context of Vapnik’s statistical learning theory [18]. The goal of the SVM algorithm is to find a hyperplane that optimally separates a higher dimensional space into different categories. SVM training consists of solving an optimization problem whose objective function gives a tradeoff between margin and misclassification error over the training dataset [19]. An advantage of the support vector method is that only a few training samples are involved in the determination of the prediction functions, facilitating the application of SVM to data mining problems with a huge amount of data. The whole formulation and some discussion can be found at [20].

SVM has been widely used in real application due to its efficient performance in machine learning problems. In the last years, different SVM methods have been successfully applied to solve the practical problems. In [21], a hybrid of k-means and SVM methods is developed and its application on breast cancer detection is presented. The k-means algorithm is applied to identify the patterns of the benign and malignant tumors which are used as features to build the dataset for SVM training. This approach achieves competitive performance results compared with other methods in cancer diagnosis. A multi-stage framework for sentiment analysis and opinion mining is proposed in [22]. This approach combines SVM and k-nearest neighbors methods, aiming to detect positive and/or negative opinion trends within weblogs containing knowledge written by baseline adopters. The authors in [23] introduced a SVM method for detection of American football head impacts using biomechanical features. A combined use of head impact sensors with video analysis was developed to the features extraction and to build training and validation datasets. A method of fault detection in wireless sensor networks based on SVM is presented in [24]. All data collected by the sensors of the network are redirected to the server that uses them to train an individual SVM with Gaussian kernel. Although this approach achieves good performance results, it requires additional communication overhead and a significant delay in data processing.

Although SVMs achieve excellent performance results, the computational time and memory requirements increase rapidly on complex and large datasets. For this reason, many research efforts have been conducted to design fast training algorithms of SVMs. The authors in [25] suggest a decomposed algorithm which divides the problem into smaller sub-problems that are solved iteratively. The method introduced in [26] proposes to reduce the size of the optimization problem by solving a sequence of sub-problems considering only a few features of the training dataset that are selected using a heuristic approach. Similarly to the aforementioned approaches, a decomposition method, called Sequential Minimal Optimization (SMO), is developed in [27]. The key idea behind the SMO method is to split the problem into the smallest possible sub-problems. Each sub-problem is solved analytically so the numerical optimization is avoided entirely, leading to a considerable reduction in computation time. More recent work [28] proposes a novel approach to select a representative subset from the training dataset using an algorithm based on convex hulls and extreme points.

### 2.2. Ensemble Learning

An ensemble of classifiers is a set of classifiers whose performance as a group improves the performance of individual classifiers. These individual classifiers are trained with subsets of the original training set and generate their own separating surfaces that will be later integrated in order to achieve more accurate and precise classification [29].

A nice theoretical property of ensembles is that the generalization error converges as the number of members of the ensemble increases. This property guarantees that overfitting will not become a problem [15]. Regarding accuracy, it can be demonstrated that an ensemble’s accuracy depends on the strength of the individual classifiers and a measure of the dependence between them. To guarantee this property, the best members of the ensemble can be chosen during the training stage.

The widely used methods for constructing ensemble learning algorithms are boosting [30] and bagging [31]. Boosting is an algorithm that works by training base learners sequentially, so in each iteration the learner assigns higher weight to the observations of the dataset that have been misclassified by its predecessor. In bagging, different sample subsets are randomly drawn from the training dataset and each subset is used to train a basic learning model in a parallel manner. To obtain the global decision of the ensemble method, the outputs of the individual models are aggregated by voting.

Ensemble learning has been successfully used in diverse applications such as text classification [32], speech recognition [33,34], sentiment analysis [35], protein folding recognition [36], or streamflow forecasting [37]. Different learning algorithms have been used as base models to build ensemble methods such as neural networks, naive Bayesian, or decision trees, among others. An ensemble method based on neural network with random weights for online data stream regression is presented in [38]. The main idea of this method is to train various neural networks with subsets of the training dataset generated from combining bootstrap sampling with random feature selection. The results indicate an accuracy improvement and reduction in computational time compared to other available algorithms from literature. In [39], an ensemble of fine-tuned naive Bayesian classifiers for text classification is proposed. A bagging method is used for ensemble construction in combination with parameter modification over learning rate and number of iterations. In [40], a novel approach for constructing ensembles of decision trees is proposed, where each tree is trained with a subset containing all features of the training set, giving a different weight to every feature. All the nodes in a tree use the same vector of random weights, but different weights are used for each tree of the ensemble.

Finally, there is extensive research that has successfully applied SVMs as base models to build ensemble methods for solving machine learning problems, often leading to improved results compared with alternative techniques. An approach developed in [41] generates a new quality training dataset through the marginal density ratios transformation on the original features. The transformed data is used to train several SVM classifiers and feed their outputs to another SVM to train the final classification model. The results show that their method performs better than other ensemble approaches in terms of accuracy and training speed. The authors in [42] compared classification performance for breast cancer prediction of an individual SVM and various SVM ensemble methods. They used bagging and boosting methods for constructing the SVM ensembles combining different kernel functions. The experimental results showed that the radial basis function (RBF) kernel SVM ensemble based on the boosting method performed better than other classifiers.

### 2.3. Voronoi Diagrams

The Voronoi diagrams are an important method of computational geometry, designed primarily for evaluating nearest neighbor over two-dimensional spatial points [43]. A Voronoi diagram is characterized by regions of proximity, making the partitioning of a plane into disjoint convex polygons where the distance of points is defined by Euclidean distance so that all points in the same polygon have the same nearest neighbor, called the centroid. Thereby, from a given polygon, every point is closer to its centroid than to any other.

The Voronoi diagrams method has been used in a wide variety of applications [44] such as virus spread analysis among mobile devices [45], cluster analysis [46,47,48], continuous location-based services [49], or high-dimensional query evaluation [50].

In recent years, several works have been published presenting novel methods in diverse fields such as computer graphics, pattern recognition, or robotics. For instance, in [51], a method to achieve cost-effective 3-D printing of stiffened thin-shell objects is proposed. For that, they use the finite element analysis to determine the regions of the object with high stress and use a given number of seeds to create a Voronoi diagram to distribute these seeds in the areas with higher stress. These seeds are mapped from a 3-D mesh to a 2-D space with least squares conformal maps (LSCMs) [52]. The authors in [53] introduce the Voronoi diagrams for the analysis of the spatial organization in team sports, such as basketball, and define the behavioral team patterns during a positional attack. The approach in [54] proposes to reduce the computation time of the robots to make quick decisions before they collide with obstacles, using Voronoi diagrams for building a roadmap in the environment of the robot.

Finally, there are numerous studies using Voronoi diagrams to tackle imbalanced classification problems [55,56,57]. These kinds of problems arise when the distribution of examples among the classes is skewed. Real-world examples abound with problems of this type from fields such as visual computing, text classification, medicine, security, finance, among others. Furthermore, in the imbalanced classification problems, the class of interest is usually the minority class (e.g., credit card fraud detection, spam detection, disease risk detection) and traditional classifiers typically maximize an overall performance, which often results in the minority class being ignored. The synthetic minority oversampling technique (SMOTE) [58] is probably the most widely used method to mitigate this problem. It is based on the generation of synthetic samples for the minority class aiming to balance the dataset. An alternative approach is that of [55]. They proposed an over-sampling method based on Voronoi regions. The underlying idea of this method is to identify exclusive regions of the feature space where the generation of new instances by random resampling provides consistent data generalization. The results of this work suggest that, in certain cases where the complexity of the datasets is high, their proposed method leads to more accurate and better classification models than using SMOTE.

## 3. pSVM Algorithm

The key idea underpinning our novel method for pSVM is to build a guided ensemble of SVM classifiers. In this ensemble, each SVM can be trained separately and in a parallel environment. The ensemble is built using a clustering method over the training set that generates a Voronoi diagram, which splits the space into a specific number of regions defined by its center. Then, these regions are used to generate the ensembles in a guided manner.

### 3.1. Data Partitioning

Typically, in a binary classification problem, a training set consisting of *n* samples can be represented as:(1)D={(xi,yi)}i=1n,
where xi∈ℝd denotes the training samples and yi∈C={−1,+1} the associated labels.

In this phase, we split *D* into *P* training subsets $D_{1},\ldots ,D_{P}$, each consisting of $n_{p}$ samples. Thus, the jth subset can be represented as:(2)$$D_{j}={\left\{\left(x_{i},y_{i}\right)\right\}}_{i=1}^{n_{j}}.$$

These subsets are created by ensuring that each $D_{j}$ maintains a similar proportion of samples from each class as in the original dataset *D*. For this, a partitioning approach separately on each of the classes $D^{C}$ is used. Thus, we can represent *D* as a collection of *C* classes:(3)$$D={\left\{D^{i}\right\}}_{i=1}^{C}.$$

The next step is to generate a Voronoi diagram from the samples of each of the classes $D^{C}$. This can be achieved using a cluster algorithm such as *k*-means [59]. The idea of this method is to find *k* regions of the space, such that any point inside its region is closer to its region’s center than to any other region’s center. It is important to remark that we do not need to find a global minimum of the optimization problem involved in the *k*-means algorithm. For our purposes, it is enough with a single execution of a limited number of iterations of the *k*-means algorithm in order to obtain regions with a balanced number of data.

In order to determine the optimal number of clusters in a dataset, several methods have been proposed [60,61,62,63]. We have adapted the Sturges rule [64] to a multi-dimensional setting. Show, given a dataset of *n* samples, the number of clusters is estimated through the formula:(4)k=1+3.332logn.

Therefore, as we mentioned above, we use *k*-means clustering on each of the classes separately leading to generate different Voronoi diagrams, one per class. Let us assume $V_{i}^{C}$ denotes the *i*th Voronoi region of class *C* and $c_{i}$ represents its associated centroid, the Voronoi diagrams of class −1 and class +1 could be represented as $V^{-}={\left\{\left(V_{i}^{-},c_{i}^{-}\right)\right\}}_{i=1}^{k^{-}}$ and $V^{+}={\left\{\left(V_{j}^{+},c_{j}^{+}\right)\right\}}_{j=1}^{k^{+}}$, respectively, where $k^{-}$ and $k^{+}$ are estimated by Equation (Equation 4).

Then, we generate the subset of all pairs, resulting as the combination of each Voronoi region of the class −1 with each of the regions of the class +1. Therefore, the new training subsets can be represented as:(5)$$\begin{array}{c} D^{\prime }\;=\;\left\{\left(V_{i}^{-},V_{j}^{+}\right)\;|\;i=1,\ldots ,k^{-}\;\mathrm{and}\;j=1,\ldots ,k^{+}\right\}. \end{array}$$

Figure 1 illustrates the steps carried out to perform data partitioning process.

### 3.2. Training

Let $D_{1},\ldots ,D_{P}$ represent the *P* training subsets generated in the previous stage that contain data samples from both classes. Then, each of these subsets can be used to train a small SVM (sub-SVM) that can be trained independently using a standard SVM training algorithm. Each of the sub-SVM will generate a sub-model.

It is important to remark that the sub-models can be perfectly trained in a parallel manner, as the input data for the sub-SVM models are independent thanks to the Voronoi partitioning. Therefore, the training subsets are distributed among all available nodes. When the number of nodes is less than *P*, several sub-SVM are trained sequentially by each node. Otherwise, each node trains a sole sub-SVM. Formally, the parallelized system composed of *N* nodes, where *H* training subsets are allocated in each node, can be represented as:(6)$$SVM^{ensemble}=\left\{SVM_{lh}|l=1\ldots N,h=1\ldots H\right\}.$$

Additionally, in order to improve training times, the number of iterations required to converge toward the solution within each sub-SVM model could be limited. This is possible because the theory underlying ensembles guarantees that the accuracy of an ensemble depends on the strength of the individual classifiers [15].

#### Learning Strategy of Each Sub-SVM

Each sub-SVM follows the typical learning strategy based on regularization theory [20]. SVMs build a classification function through the solution of the following optimization problem:(7)minf∈Hk1n∑i=1nLyi,f(xi)+Mfk2,
where (xi, yi), i=1,…,n, is a training dataset with xi∈ℝd and yi∈{+1,−1}; HK is a reproducing kernel Hilbert space (RKHS) with a kernel *K*; fK is the norm of *f* in the RKHS; Lyi,f(xi) is a loss function; and the cost M>0 is a constant that penalizes non-smoothness of the possible solutions to optimization problem (Equation 7). The SVM loss function for classification purposes is:(8)Lyi,f(xi)=max1−yi×f(xi),0.

It can be shown that the solution of problem (Equation 7) using Equation (Equation 8) leads to a smooth function f*∈HK, such that:(9)f*(x)=∑i=1nαiK(x,xi)+b*,
where αi and b* are constants; K(x,y)=ϕ(x)Tϕ(y) is the kernel function that generates HK; and ϕ:ℝn→ℝp is a mapping defining *K*. ϕ maps the data from ℝd (known as the “input space”) into ℝp (the so-called “feature space”).

The main steps of the training algorithm are illustrated in Algorithm 1.
**Algorithm 1:** pSVM training algorithm.**Data**: $D={\left\{\left(x_{i},y_{i}\right)\right\}}_{i=1}^{n}$, $x_{i}\in R^{d}$, $y_{i}\in C=\left\{-1,+1\right\}$;*N*: number of nodes;**Result**: S (set of Voronoi regions pairs);$SVM^{ensemble}$ (ensemble of SVM);_1_ ${\left\{D^{c}\right\}}_{c=1}^{C}\leftarrow$ build a collection of *C* classes;_2_ $k^{c}=1+3.332\log n^{c}$, where $n^{c}$ is the number of samples in $D^{c}$ and c∈C;_3_ $V^{c}={\left\{\left(V_{i}^{c},c_{i}^{c}\right)\right\}}_{i=1}^{k^{c}}\leftarrow$*k*-means($D^{c}$, $k^{c}$), where $c_{i}^{c}$ is the centroid of the Voronoi region $V_{i}^{c}$ and c∈C;_4_ $D^{\prime }\leftarrow D_{ij}^{\prime }=\left\{\left(V_{i}^{-},V_{j}^{+}\right)|i=1,\ldots ,k^{-1}\mathrm{and}j=1,\ldots ,k^{+1}\right\}$;_5_ $H\leftarrow \frac{length(D^{\prime })}{N}$;_6_ $S=\{S_{lh}|l=1\ldots N,h=1\ldots H\}$, where S is the distributed version of $D^{\prime }$ among the nodes;_7_ **for** *h←1 to H***do**;_8_ ∣($SVM_{lh}=train$-$SVM(S_{lh})$, l=1,…,N_9_ $SVM^{ensemble}=\left\{SVM_{lh}|l=1\ldots N,h=1\ldots H\right\}$

### 3.3. Classification

Once the training phase is finished, an ensemble of sub-SVMs could be used to classify new data. Instead of using all sub-SVMs, the proposed algorithm selects a subset of them based on *k* nearest neighbor approach (*k*-NN) [65]. To achieve this, for each new individual, the Euclidean distance with the centroids of the Voronoi regions is computed and the γ closest ones of each class are selected. Let $T^{-}$ and $T^{+}$ represent, respectively, the γ nearest Voronoi regions of class −1 and class +1 to the new individual. Then, a subset of the training subsets of Equation (Equation 5), $T\subset D^{\prime }$, is generated as the Cartesian product of $T^{-}$ and $T^{+}$:(10)$$\begin{array}{c} T\;=\;\{\left(v^{-},v^{+}\right),\;v^{-}\in T^{-}\;\mathrm{and}\;v^{+}\in T^{+}\}. \end{array}$$

Thereby, only the sub-SVM trained with the subsets on *T* are taken into account for prediction, discarding the remaining sub-SVM.

The pSVM uses a voting scheme similar to the one described in [66], where each new individual is evaluated by the selected sub-SVM, being the evaluation provided by each sub-SVM considered as a vote. Once all the votes are aggregated, the new individual is classified as a member of the most voted class. If there is an even number of sub-SVMs, ties during the voting of some individuals might take place. Those individuals are assigned at random, although more sophisticated schemes may classify those individuals as undetermined in order to evaluate their classification later by an expert. To be more specific, if *t* sub-SVMs are available, the class assigned to an individual *z* will be denoted as class(z) and determined by Equation (Equation 11).
(11)$$class\left(z\right)=\left\{\begin{array}{cc} sgn\left(\sum_{i=1}^{t}prediction_{i}\left(z\right)\right) & \mathrm{if}\sum_{i=1}^{t}prediction_{i}\left(z\right)\neq 0\\ \pm 1(randomly) & \mathrm{if}\sum_{i=1}^{t}prediction_{i}\left(z\right)=0 \end{array}\right.$$
where $prediction_{i}\left(z\right)$ is the vote corresponding to sub-SVM *i* and sgn is a function defined as:(12)$$sgn\left(x\right)=\left\{\begin{array}{cc} 1 & \mathrm{if}x>0\\ 0 & \mathrm{if}x=0\\ -1 & \mathrm{if}x<0 \end{array}\right.$$

The steps to perform the classification stage are summarized in Algorithm 2.
**Algorithm 2:** pSVM classification algorithm.
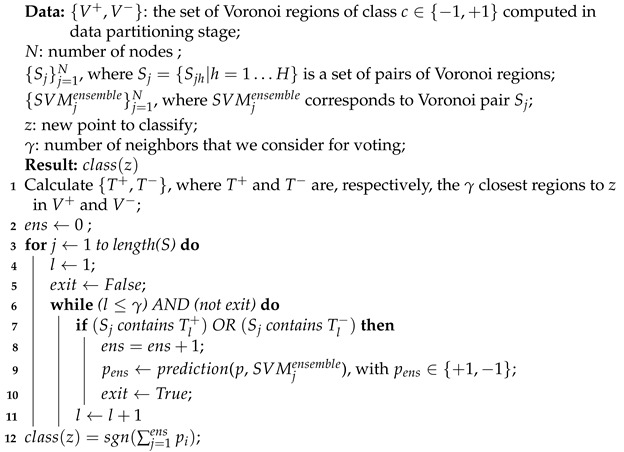


### 3.4. Computational Complexity

The following theoretical result shows that the computational complexity of our proposal lowers the computational complexity of a single SVM.

**Theorem** **1.**
*For a bounded number of iterations of the k-means method, the worst case computational complexity of the pSVM training algorithm proposed in this work amounts to $O({\left(\frac{n}{\log n}\right)}^{3})$.*


**Proof.** The worst case computational complexity of using a single SVM is $O(n^{3})$ [67]. Regarding the *k*-means algorithm, it is well known that the optimization problem involved in this method is NP-hard [68]. In practice, truncated versions of this algorithm are used, so that a rough worst case bound can be assumed to be O(I∗k∗n), where *I* is the number of iterations and *k* is the number of Voronoi regions. In a typical truncated version of *k*-means method, the maximum number of iterations is fixed. Therefore, for a large *k*, this computational complexity can be considered to be lower than $O(n^{2})$. By construction, each sub-SVM used in the pSVM algorithm has a computational complexity of:
$$O\left({\left(\frac{n}{1+3.332\log n}\right)}^{3}\right)=O\left({\left(\frac{n}{\log n}\right)}^{3}\right).$$Since each sub-SVM can be trained simultaneously to the rest, the overall computational complexity of our pSVM algorithm amounts to $O\left(n^{2}\right)+O\left({\left(\frac{n}{\log n}\right)}^{3}\right)$, that is, $O({\left(\frac{n}{\log n}\right)}^{3})$.    □

## 4. Experimental Results

In this section, we provide empirical evidence of our analysis of guided pSVM based on Voronoi regions using two synthetic datasets and discuss the results. All experiments were conducted on a workstation running Linux with two Intel Xeon E5-2630 (6 cores per CPU, two threads per core), at 2.3 GHz and 64 GB of RAM memory. A prototype implementing the algorithms described in Section 3 was developed using the statistical software R v3.6.0 [69], RStudio v1.4.1106, and the following additional R packages. Data processing was carried out with packages stats v3.4.4 and dplyr v1.0.0. Visualization was undertaken using package ggplot2 v3.3.3. We created a custom function based on package e1071 v1.6-8 to build the SVM classifier, so that we can limit the number of iterations to achieve convergence. Finally, parallelization was carried out through package doParallel v1.0.16.

### 4.1. One Region with Two Partially Overlapping Classes

A first simple experiment consists of the classification of two partially overlapping classes, where all the data are located in the same space region. Figure 2 shows the situation for the two-dimensional case. We use two *d*-dimensional Gaussian distributions (x,y)∼Nm(μm,σm), m∈{1,2} to simulate each class, where μ1=(0,0), μ2=(2,2) and the covariance matrix is ([1,0];[0,1]) for both distributions. In particular, μ1=(0,…,0)∈ℝd and μ2=(2,…,2)∈ℝd. The covariance matrices σm were randomly generated with diagonal (1,…,1),∈ℝd. The experiment is executed for d=2. A balanced dataset is artificially generated by randomly sampling 500,000 training points and 50,000 testing points from each class.

As each class has a sample size of 500,000 points, from Equation (Equation 4), the number of clusters obtained is 45 ($k^{-1}=k^{+1}=45$). Since the Voronoi diagrams corresponding to classes −1 and +1 are very similar, for conciseness, in Figure 3 we only show the diagram for class +1.

#### Results

Here, we evaluate the performance of pSVM versus a single SVM and a standard SVM ensemble [70]. We choose the well-known SVM implementation provided by the libSVM library [14]. We randomly split each dataset into a training and a testing group, where the training set is 10 times larger than the testing set, and run all methods using this setup. This procedure is repeated 10 times and we obtain the average value and standard deviation of the accuracy performance measure, that is, the fraction of individuals correctly classified, given by:$$accuracy=\frac{t_{p}+t_{n}}{t_{p}+f_{p}+t_{n}+f_{n}},$$
where $t_{p}$ (true positives) are defined as the set of individuals correctly classified in a certain class, $t_{n}$ (true negatives) as the set of individuals correctly left out of a certain class, $f_{p}$ (false positives) as individuals incorrectly classified in a certain class, and $f_{n}$ (false negatives) as individuals that have been incorrectly left out of a certain class. Because we are using a balanced dataset, this measure will work correctly providing reliable information to assess the performance of these methods.

Each method is run with two different kernel functions (see [20] for different choices), namely a linear kernel and a radial basis function (RBF) kernel with parameters estimated by cross validation. Then, we compare the following approaches:Single SVM, ensemble, and pSVM with no limit of iterations;Single SVM, ensemble, and pSVM with a limit of 10 iterations;Single SVM, ensemble, and pSVM with a limit of 1 iteration.

As mentioned in Section 3.3, a *k*-NN approach based on Voronoi regions is used to select the sub-SVMs considered as classifiers. It seems obvious that different values of *k* lead to different performance results. To select the optimal value of this parameter empirically, we tested different choices for *k*: 1, 3, 5, 7, and 9. As Figure 4 shows, the accuracy improves while we increase *k* from 1 to 7, whereas it is relatively stable for *k* larger than 7. Therefore, k=7 was chosen as the optimal number of Voronoi regions used in the classification scheme.

Table 1 shows the average classification accuracy and the standard deviation of the algorithms for ten runs on the synthetic dataset. As we can see, when the number of iterations required to converge to the solution is not limited, all approaches provide accuracy results over 91.0%. Best results are obtained by the ensemble with RBF kernel and the pSVM with RBF kernel being, respectively, 92.69% and 92.65%. However, when the number of iterations is limited, for energy saving reasons, the only methods providing consistent results are the two versions of the pSVM approach, which do not seem to be affected by the iteration limit. In these cases, the best accuracy results are 92.65% and 92.57%, for the two limited versions of the pSVM with RBF kernel. Furthermore, it is important to notice that the only method whose accuracy systematically remains over 91.0% is pSVM, for all versions.

### 4.2. Eight Multi-Dimensional Regions with Two Partially Overlapping Classes

This second experiment is based on a synthetic dataset that emulates a federated network of sensors. As mentioned above, such networks are characterized by providing data distributed in different regions in which it is necessary to categorize events in different classes. For this experiment, to simulate each class we generate sixteen *d*-dimensional Gaussian distributions $\left(x,y\right)∼N_{m}\left(\mu_{m},\sigma_{m}\right)$, m∈{1,…,16}, paired two by two. For simplicity, 16,000 elements in a 10-dimensional space have been generated (1000 elements per class for each region), although similar results were obtained for larger dimensional settings and datasets, up to one million elements. Figure 5 depicts this dataset for d=2.

#### Results

Again, on this dataset, we compare the performance of our pSVM approach to a single SVM and a standard SVM ensemble. For the three methods, two different versions are implemented: one using a linear kernel and another using an RBF kernel, with parameters estimated by cross validation. Moreover, for the ensemble and pSVM approaches, different classification schemes are used. In particular, for both methods, we implement the classification scheme described in Algorithm 2 for different values of the γ parameter, namely: γ=1, γ=7, and γ=15. In the case of the single SVM, the classification scheme is made up of a single decisor and, in Table 2, the result appears in the row corresponding to γ=1. In a similar manner to the previous example, we randomly split each dataset 10 times into a training and a testing set. Similarly, we run the methods and calculated the average value and standard deviation of the accuracy performance measure.

Table 2 presents the average classification accuracy and the standard deviation of the algorithms for ten runs on the multidimensional dataset. As we can observe, the best result for the linear kernel versions of the algorithms are always provided by the pSVM approach. This is because the method has been specifically designed for data whose structure is similar to that of a federated network of sensors. As expected, using a more complex kernel, the ensemble approach improved its results, especially for large values of γ. Unfortunately, this approach requires a cross-validation process to estimate the parameters of the kernel, whereas the linear kernel does not require this additional step. Finally, it is remarkable that, under a severe reduction in the number of training iterations up to a single one, the best overall accuracy result (94.95%) is obtained by the pSVM with an RBF kernel.

### 4.3. A Numerical Estimation of Training Time

Finally, for completeness, we provide a table with the execution time exhibited by the different methods on the 10-dimensional example in Section 4.2. It is important to notice that, although the smallest time results are obtained by the single SVM with a limited number of iterations, these implementations provide very poor classification results. Therefore, it would never be chosen in practice. Considering a tradeoff between accuracy and training times, the best implementations correspond to the pSVM approach with linear kernel. In particular, the pSVM version without a limit of iterations is, on average, up to 11.88 times faster than the single SVM with linear kernel. This magnitude is in accordance with the expected proportional reduction in the order of log(n), shown in Section 3.4.

Table 3 summarizes the execution time (in seconds) for all versions of the methods implemented in this comparative.

## 5. Conclusions

In this paper, we present a novel method for accelerated training of parallel Support Vector Machines that is especially well-suited for problems involving a federated network of sensors where optimization of energy consumption is required. The proposed algorithm builds on a parallel training alternative of SVM ensembles (pSVMs), determined by Voronoi regions. Experimental results indicate that training time is reduced according to the analytical computational complexity analysis of the method. This method exhibits a stable performance when the convergence iterations within the training stage are limited. In particular, it is important to remark that the simplest version of this pSVM approach, that is, the one using a linear kernel, makes this method the most appropriate for a parallel implementation. In that case, the evaluation of the kernel function simply involves a scalar product without additional parameters, and thus cross validation is not needed.

Concerning further research, a more detailed complexity analysis including the effect of the dimension of the data may be interesting, especially for data coming from very high dimensional settings. Another interesting area of future research is the development of multiclass versions of the pSVM approach. As well, a drastic acceleration of the training stage could be achieved through a hardware implementation of this novel approach. To this aim, pSVM versions with a limited number of iterations are even more suitable.

Regarding possible shortcomings of our proposal, there is still room for improvement. Alternatives for constructing the Voronoi regions should be explored. Another limitation that requires future attention is that the Sturges formula was originally developed for one-dimensional data. Therefore, it would be advisable to develop a more sophisticated version, including in its closed-form the dimension *d* of the representation space. This is related to the necessary compromise between the number of Voronoi regions and the number of data elements comprised in each region, which may be crucial to improve the performance of this method.

## Figures and Tables

**Figure 1 entropy-23-01605-f001:**
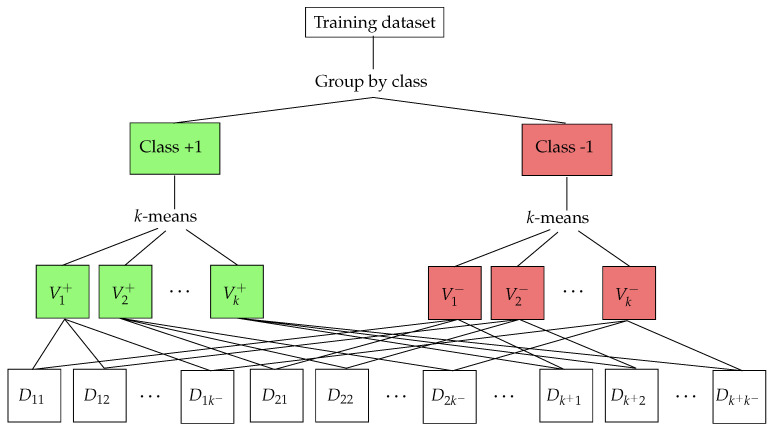
The flowchart of data partitioning method.

**Figure 2 entropy-23-01605-f002:**
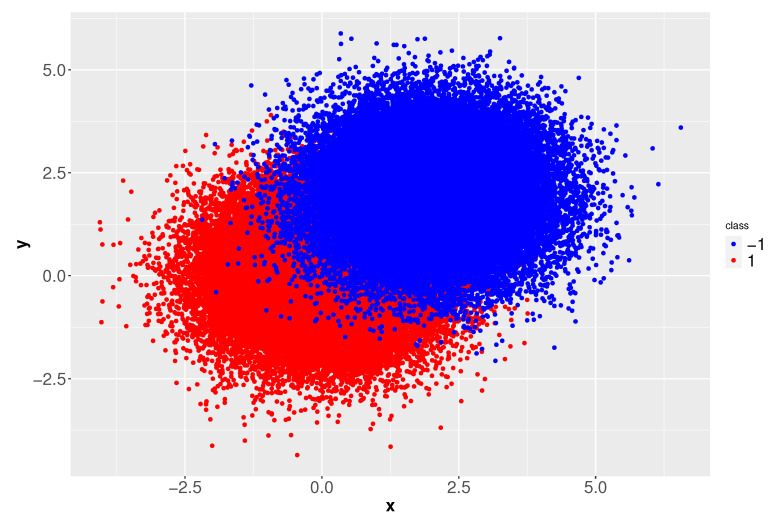
An example of the synthetic dataset in a 2-D feature space.

**Figure 3 entropy-23-01605-f003:**
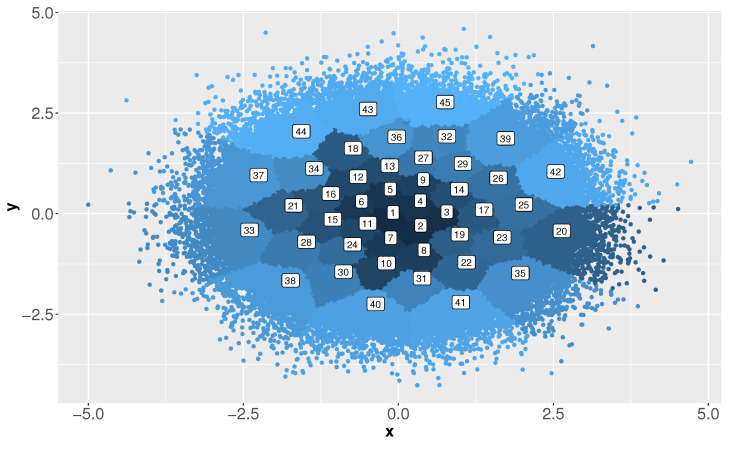
Voronoi diagram for class 1.

**Figure 4 entropy-23-01605-f004:**
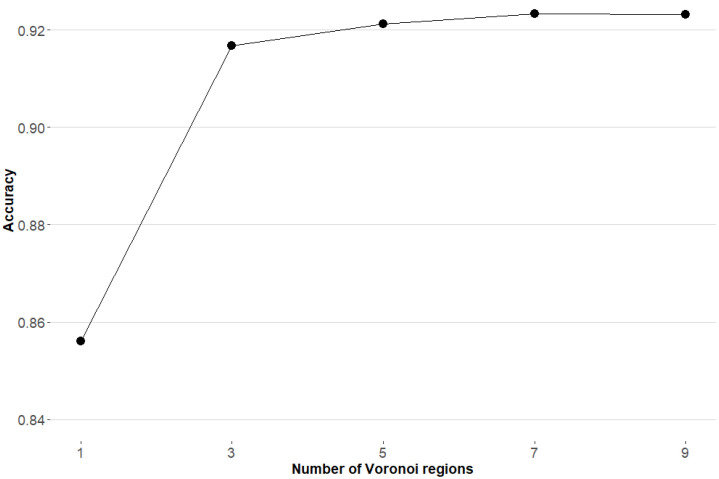
Number of Voronoi regions, for each class, selected for classification.

**Figure 5 entropy-23-01605-f005:**
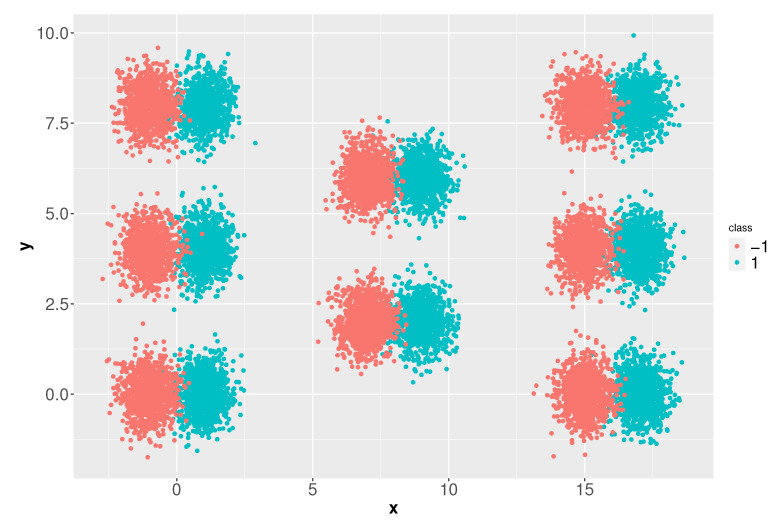
A two-dimensional example with two classes and eight regions.

**Table 1 entropy-23-01605-t001:** Average (standard deviation) for accuracy for each method. The method with the best accuracy is boldfaced.

Iterations	SVM (Linear Kernel)	SVM (RBF Kernel)	Ensemble (Linear Kernel)	Ensemble (RBF Kernel)	pSVM (Linear Kernel)	pSVM (RBF Kernel)
No limit	0.9223 (0.0030)	0.9246 (0.0125)	0.9237 (0.0011)	0.9269 (0.0076)	0.9130 (0.0139)	0.9265 (0.0122)
10	0.6641 (0.1929)	0.4465 (0.1226)	0.8963 (0.0128)	0.4958 (0.0227)	0.9150 (0.0049)	0.9265 (0.0120)
1	0.6543 (0.2790)	0.4241 (0.1266)	0.8887 (0.0167)	0.3107 (0.0189)	0.9107 (0.0078)	0.9257 (0.0129)

**Table 2 entropy-23-01605-t002:** Average (standard deviation) for accuracy for each method. The method with the best accuracy is boldfaced.

Iterations	γ	SVM	SVM	Ensemble	Ensemble	pSVM	pSVM
		(Linear Kernel)	(RBF Kernel)	(Linear Kernel)	(RBF Kernel)	(Linear Kernel)	(RBF Kernel)
No limit	1	0.6183 (0.0580)	0.9747 (0.0047)	0.4404 (0.1000)	0.9687 (0.0028)	0.9641 (0.0053)	0.9695 (0.0051)
	7	-	-	0.4033 (0.0100)	0.9751 (0.0036)	0.8493 (0.0563)	0.8297 (0.0219)
	15	-	-	0.391 (0.0029)	0.9763 (0.0042)	0.5437 (0.0799)	0.6566 (0.0931)
10	1	0.5730 (0.0025)	0.5506 (0.0100)	0.4779 (0.0097)	0.6513 (0.0183)	0.8970 (0.0249)	0.9495 (0.0073)
	7	-	-	0.4289 (0.0083)	0.8956 (0.0088)	0.8218 (0.0787)	0.7950 (0.0670)
	15	-	-	0.3910 (0.0029)	0.9555 (0.0074)	0.5220 (0.0917)	0.6714 (0.1103)
1	1	0.5350 (0.0399)	0.5421 (0.0138)	0.4276 (0.0077)	0.5570 (0.0157)	0.7675 (0.0154)	0.8331 (0.0111)
	7	-	-	0.4372 (0.0099)	0.6736 (0.0093)	0.7637 (0.0449)	0.7616 (0.0288)
	15	-	-	0.4303 (0.0156)	0.7600 (0.0151)	0.5329 (0.0400)	0.6639 (0.0153)

**Table 3 entropy-23-01605-t003:** Average (standard deviation) for training time. The method with the shortest training time is boldfaced.

Iterations	SVM (Linear Kernel)	SVM (RBF Kernel)	Nodes	Ensemble (Linear Kernel)	Ensemble (RBF Kernel)	pSVM (Linear Kernel)	pSVM (RBF Kernel)
			4	15.2576 (0.2009)	7.9840 (0.0817)	3.0220 (0.1934)	3.7600 (0.2626)
No limit	29.8763 (6.9040)	5.0406 (0.0184)	9	9.1846 (0.4014)	5.3840 (0.1412)	2.5140 (0.1260)	2.9566 (0.1526)
			16	8.6923 (0.1162)	4.4406 (0.0155)	2.6280 (0.2912)	2.6910 (0.0818)
			4	2.5753 (0.0307)	2.8840 (0.1424)	2.3566 (0.1353)	2.7400 (0.1582)
10	0.1240 (0.0006)	0.1202 (0.0015)	9	1.9853 (0.0186)	2.2166 (0.0200)	1.8656 (0.0558)	2.0940 (0.1471)
			16	1.8790 (0.1065)	2.0703 (0.1079)	1.7970 (0.1799)	1.9126 (0.0489)
			4	2.4300 (0.1455)	2.5233 (0.1459)	2.4100 (0.2364)	2.5756 (0.0895)
1	0.0933 (0.0011)	0.0683 (0.0049)	9	1.8416 (0.0256)	2.1013 (0.2426)	1.8203 (0.0592)	1.8686 (0.0499)
			16	1.8903 (0.0584)	1.9260 (0.0270)	1.8460 (0.1292)	1.8860 (0.0770)

## Data Availability

Not applicable.

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
