# Peer review of "Toward Accelerated Training of Parallel Support Vector Machines Based on Voronoi Diagrams"

_entropy, 2021, doi:10.3390/e23121605_

Round 1

Reviewer 1 Report

1). The method is described only in two pages out of ten pages. The method should be more clear and more analytically presented.

2). While in the title the authors use the word “Accelerated Classification”, they did not present any computational complexity analysis or at least computational time values. Analysis of the computational complexity shoud be reported. I think that this is vital in order to support their claim.

3). The training process is set up to perform the parallel training of the SVM modules. However, the exact learning strategy is not clear.

4). Also, the classification module is small in size, despite the fact that along with the training process constitute the very core of the method.

5). The main contributions of the algorithm are not clear. As far as I see, the authors just apply some steps, that exist in the literature, in sequence. However, the main contributions of the proposed strategy in comparison to other similar approaches should be described in detail. Please note that the idea of SVM ensemble is not new.

6). The data set used for experimentation is trivial. There is no reason to use GPU for such small data set. It may have large number of points, but these are just 2-dimensional points. A typical CPU could manage to handle that size easily. Therefore, the authors need to test their method in more complex and larger data sets. In addition, it lacks a statistical comparison with standard statistical tests, where the method must be compared to at least two other similar approaches.

Author Response

We are very grateful to your comments, suggestions and discussion, that we have added as follows.

Reviewer 2 Report

1. The subsections 1.1 and 1.2 should be included in a "Related works" section. Your paper should involve the related works section.

2. The overall contents in section 1.1 and 1.2 are insufficient to show background knowledge about the ensemble learning and support vector machine. More detailed explanations about them are required.

3. In Equation 7, what is the function sign( * )? Its detailed explanation should be involved.

4. In page 5, the title of Section 3 (i.e., "Experimental setting") is inadequate as its section title. This section shows not only the experimental configurations but also their experimental results. Thus, this section's title should be modified as "Experiments" or "Experimental results."

5. In Figure 2, 3, and 4, it is good to add the axis titles of x-axis and y-axis to further enhance their readability.

6. What are the parameters of which your proposed method is composed? Were their sensitivity tests conducted in your experiments? How are the parameters determined? In the experiment section, more detailed explanations about them should be shown in your revised paper.

7. What are the shortcomings or weaknesses of your proposed method? To increase the proposed method further, what is the your future study plan? In the conclusion section, it is good to show them in detail.

8. More recent papers (conference or journal papers published in 2019 - 2021) should be added in your reference list. The current reference lists are too old to show the state-of-the-art study issues sufficiently.

Reviewer 3 Report

The paper is devoted to the development of the classification algorithm, which is based on the ensemble of SVM classifiers for partitioned data.  The novelty of the work which differs it from the existing ensemble SVM is the way of partitioning the training dataset. Namely, for this purpose, the authors apply the Voronoi diagram. The paper is well-written, the results are supported with numerical examples.  However, I have a few essential comments concerning evidencing the results and presentation of the work.

Comments:

  1. In my mind, the usage of Voronoi diagrams is the keystone of the algorithm offered. There are other works related to machine learning, in particular, to classification tasks, which use Voronoi diagrams. Your introduction should be improved with the help of reviewing recent works on Voronoi diagrams, especially their application for machine learning problems.
  2. I think that the title of the paper should be changed. It is not reflecting the results of the work. What is the role of the Internet of Things in your work to include it in the title of the paper? On the other hand, I think that Voronoi diagrams are more important results to be reflected in the title. For example, “Towards Accelerated Classification with Parallel Support Vector Machines Based on Voronoi Diagram”
  3. One of the most important results of your work is to show that pSVM requires less training time. The only evidence is that “The reason for reducing training time in pSVM lies in the fact that few iterations are needed to find an optimal solution” and numerical experiment. Why not present the computational complexity of the problems clearly. For example, the computational complexity of the Voronoi diagram construction is O(NlogN), for SVM is O(N^2) or O(N^3) (or like this), etc. Therefore you can formulate the estimate of the time which is required for the training analytically. Then you can evidence it with the help of a numerical experiment.
  4. The same is concerning your conclusions that “training time is reduced by 7-12x”. It is only the experiment. You should explain it clearly in the terms of computational complexity. It is crucial for your paper.
  5. pSVM algorithm should be presented properly. Fig. 1 does not clearly present the algorithm. I recommend presenting pSVM clearly as an algorithm or flowchart, indicating what is input data, basic steps such as the choice of $k$, constructing Voronoi diagram, training, etc., output data. That is present 2.1, 2.2, 2.3 clearly using variables and operations. Currently, Fig. 1 is too general a presentation of the algorithm.
  6. page 5, lines 174-175. You write “… we have randomly split each dataset into a training group and a testing group. For each dataset, we have done these random divisions 10 times.” Do you mean the resampling strategy used? If so, it is better to call it 10-fold cross-validation.
  7. Conclusions should be improved with the help of adding some discussion and open problems. For example, ability to apply pSVM in the case of multiple dimensions or Big Data? How to cope with computational complexity in such cases. Unfortunately, the question of the Internet of Things has been left undiscovered in the paper.
  8. Please check grammar. page 5, line 175: "we have done these random division" (divisionS)

Round 2

Reviewer 1 Report

the authors have substantially improved the manuscript.
Therefore I suggest to accept the paper as it is.

Reviewer 3 Report

All my comments were reflected in the revised version of the manuscript. For the reason given, I recommend the manuscript for publication